# CRISPR/Cas9—A Promising Therapeutic Tool to Cure Blindness: Current Scenario and Future Prospects

**DOI:** 10.3390/ijms231911482

**Published:** 2022-09-29

**Authors:** Irshad Ahmad

**Affiliations:** 1Department of Bioengineering, King Fahd University of Petroleum and Minerals (KFUPM), Dhahran 31261, Saudi Arabia; irshad@kfupm.edu.sa; Tel.: +966-13-8608393; 2Interdisciplinary Research Center for Membranes and Water Security, King Fahd University of Petroleum and Minerals (KFUPM), Dhahran 31261, Saudi Arabia

**Keywords:** CRISPR, retinal degeneration, eye diseases, blindness, eye therapeutics, viral vectors, non-viral vectors

## Abstract

CRISPR-based targeted genome editing is bringing revolutionary changes in the research arena of biological sciences. CRISPR/Cas9 has been explored as an efficient therapeutic tool for the treatment of genetic diseases. It has been widely used in ophthalmology research by using mouse models to correct pathogenic mutations in the eye stem cells. In recent studies, CRISPR/Cas9 has been used to correct a large number of mutations related to inherited retinal disorders. In vivo therapeutic advantages for retinal diseases have been successfully achieved in some rodents. Current advances in the CRISPR-based gene-editing domain, such as modified Cas variants and delivery approaches have optimized its application to treat blindness. In this review, recent progress and challenges of the CRISPR-Cas system have been discussed to cure blindness and its prospects.

## 1. Introduction

The eye is an important mammalian organ with a complex anatomical structure that is produced and regulated by different types of multiple ocular tissues. The disruption of any ocular tissue can cause visual impairment and vision loss, which can significantly influence the quality of life in mammals, including human beings. Damaging of the retina and optic nerve are the major causes of vision loss due to aging. Recent data from WHO shows that 1.3 billion people are affected with vision impairment among which the majority of people are above the age of 50 years. Age-related muscular degeneration (AMD), cataract, dry eye, and glaucoma are the common types of ocular diseases in the world [1]. During the past two decades, a lot of research work has been done to discover different molecular therapeutic tools to treat ocular diseases. These strategies include gene therapy through viral delivery methods, implants, hydrogels, liposomes, and nanoparticles-based delivery to repair ocular diseases. However, the mentioned strategies have major drawbacks, such as immune rejections, and biocompatibility issues of hydrogels, implants, and nanoparticles.

The recently developed gene-editing technology, clustered regularly interspaced short palindromic repeats (CRISPR/Cas9) is revolutionizing every field of biological science research as well as ophthalmology. The CRISPR/Cas9 is a precise gene-editing technology that has emerged as a novel therapeutic tool to treat ocular diseases by restoring the vision in human beings. CRISPR/Cas9 was discovered in 1987 and was reported for the first time in 2013 for editing the mammalian genome. Consequently, it has been used as a therapeutic agent to treat genetic diseases such as retinal dystrophies, neurodegenerative diseases, etc. There are three classes of CRISPR-Cas system, I to III. Each class of Cas system has been divided into the CRISPR subtypes [2]. The mechanism is very simple, designing the guide RNA (gRNA) according to any target gene and then combining it with Cas9 to form the ribonucleoprotein (RNP) (Figure 1). The RNP can be directed by gRNA to cut the specific nucleotide sequence of DNA that triggers the DNA repair system, non-homologous end joining (NHEJ). The NHEJ DNA repair system is not efficient and could lead to small insertion or deletions (INDELS) that ultimately knock-out (KO) the gene of interest. It may cause the frameshift mutation in the particular gene aimed to be knocked out in the specific genome. Similarly, if the donor template has been provided with an insert of any foreign gene or to correct the single-nucleotide polymorphism (SNP), that leads to a DNA repair system e.g., homologs direct repair system. In this case, the provided donor single-stranded DNA can be used as a template DNA to repair the target DNA and ultimately knock-in (KI) the gene of interest. These strategies make the CRISPR/Cas9 system less costly as compared to conventional gene-editing tools [3].

Rapid advancement in the genome editing field has led to the development of precise base and prime editing technologies. Due to favorable anatomical eye structure and privileged immunogenic characteristics, inherited retinal diseases are at the forefront of CRISPR/Cas-based therapies. Genome-wide screening and CRISPR/Ca-based gene knockout techniques can be used to investigate eye disease mechanisms for the development of novel therapies. Leber congenital amaurosis type 10 is a genetic eye disease developing in children due to a mutation in the CE9290 gene that causes the loss of CEP290 protein and gradual loss of eyesight. Recently it has been fully cured by fixing the mutant gene through the CRISPR gene knock-in (KI) strategy. An ongoing CRISPR/Cas9-based clinical trial (NCT03872479) to cure LCA is in phase III and has been sponsored by Editas Medicine, Inc. In this research CRISPR/Cas9 has been used to correct the CEP290 gene and in near future, it will be approved to treat LCA [4]. Herpes Simplex virus (HSV) keratitis is a viral infection of the cornea that causes blindness. Worldwide, an estimated 491 million people aged 15–49 are affected by HSV. A CRISPR/Cas9-based clinical trial (NCT04560790) to treat HSV keratitis diseases is in phase II and has been sponsored by Shanghai BDgene Co., Ltd., Shanghai, China. In this study, CRISPR/Cas9 has been employed to disturb the *UL8/UL29* genes to inhibit the growth of HSV [5,6]. This article reviews and highlights the CRISPR/Cas9 as an emerging therapeutic tool for the treatment of genetic eye diseases. Its potential applications have been elaborated to cure blindness in human beings and other mammalian species.

## 2. Genetic Causes of Retinal Degeneration and Blindness

The current development in gene therapy research has played an important role in understanding the genetic origin of structural and functional defects in common human eye diseases for instance age-related macular degeneration (AMD), glaucoma, cataract, myopia, Stargardt’s disease (SD), retinitis pigmentosa (RP), Marfan syndrome (MFS), polypoidal choroidal vasculopathy (PCV), uveal melanoma (UM), and optic atrophy. Table 1 shows the gene variants associated with the mentioned eye diseases.

AMD commonly occurs in old people that causes the loss of central vision as a result of macular degeneration which is the vital area of the retina [7,8,9]. Multiple SNPs occur in many genes such as *NOS2A* which increases the likelihood of AMD, especially SNPrs8072199 [10]. Recently 3 SNPs in the *MMP-9* locus (*rs4810482*, *rs17576*, and *rs17577*) have been reported that are highly associated with an increased risk of AMD [11]. Moreover, *TIMP-3* risk alleles and mutations are directly linked with retinopathies such as AMD and SFD [12,13]. Another study has reported the association of *HTRA1* and *ERCC6* with AMD [14]. Researchers have reported that *CALM2* might be the cause of glaucoma since it takes part in the death of retinal ganglion cells (RGC) that disturbs the cell communication system. Moreover, the *MPP7* was found to be involved in primary open angle glaucoma (PAOG) through its dysfunction [15]. Additionally, multiple defective genetic regions have been identified that are related to glaucoma such as *CYP1B1*, and optineurin. *LOX1* was also recognized to be one of the causes of exfoliation type glaucoma. Early-onset glaucoma can be due to mutations in *PAX6*, *FOXC1*, *MYOC*, *LTBP2*, and *PITX2*. Finally, gene testing can help in recognizing people who can participate in clinical trials for the treatment of glaucoma [16,17]. It was discovered that inherited cataract is affiliated to more than 50 loci however the genetics of many cataracts are still unidentified. Multi-gene panel have been used to study these mutations such as *GEMIN4* mutation in cataract individuals. A parent mutation has been discovered in *RIC1* that exists in patients with cataract and by sequencing of the cataract gene it was exposed as a truncating mutation in *TAPT1* [18]. Recently, it was reported that more than 10 mutations can occur in *MIP* which is associated with cataracts [19,20,21]. Myopia is a short-sightedness where light is concentrated in front of the retina rather than on the retina. It was identified that *HGF* and *C-MET* are associated with myopia in Asians but mutations in the *C-MET* receptor in Asians were not reproduced in Caucasians [22]. Further studies showed that mutation in *UMODL1* may have a role that contributes to the susceptibility of myopia. Moreover, it was discovered that a functional SNP at 3′UTR of *PAX6* can contribute to the increased risk of myopia [23]. Stargardt’s disease (SD), is shortsighted due to the fatty material accumulation on the macula of the eye. There are more than 800 mutations identified in *ABCA4* that are associated with SD [24]. In one report two distinct phenotypes (LCA and SD) have been identified to cause severe retinal degeneration. The cases with LCA and SD were due to mutation in *CRB1* and *ABCA4,* respectively [25]. Retinitis pigmentosa (RP) is, a disorder that affects the light-sensitive layer of retina, and the patients exhibit gradual loss of sight. There are more than 250 mutations identified for 100 genes but only 50% of the defected have been identified [26,27,28,29]. A higher proportion of RP2-mediated XLRP has been reported in Danish people [30]. After extensive studies, it was reported that two rare and deleterious variants *p. Arg281Cys* and *p. Arg487** were discovered in *AGBL5* [31]. Marfan syndrome (MFS), caused by genetic mutations in the connective tissues such as mutations in *FBN1* on chromosome 15 have been discovered in patients which can lead to many abnormalities that includes increased TGF-β1 causing irregularity in signaling [32,33,34,35]. SNPs that can change the *TGFBR2* are connected to MFS [36]. Polypoidal choroidal vasculopathies (PVC) can be genetically separated into polypoidal CNV and typical PCV. The gene variants *C2* and *CFB* have been revealed to be associated with polypoidal CNV [37]. Other studies have reported the association of *C3*, *SERPING1* and *PEDF* with PCV [38,39]. Uveal melanoma (UM), caused due to *BAP1* suppressor mutations that are linked to the development of different tumors in pigment cells [40]. Hence, the roles of *CDKN2A*/*P16INK4A*, *CDK4*, and *P14ARF* germline mutations were tested. In addition, the *BRCA1/2* germline mutations and the patient history were also evaluated in the monitoring of UM [41,42]. Inherited optic neuropathies is the most typical inherited form of vision loss due to an injury in the mitochondria affecting young males. The greater number of cases with this disorder is because of a three-point mutation in mtDNA affecting complex I or *ND* genes (*G11778A*, *G3460A* and *T14484C*), while dominant optic neuropathy (DOA) is a non-sex chromosomal hereditary optic nerve neuropathy that is the most widely known inherited neuropathy. Despite the fact that it is heterogenous, a main locus has been mapped to chromosome 3q28 with a mutation in *OPA1* [43,44]. Gene-editing methods such as RNAi and CRISPR/Cas9 technology can be used to correct the mentioned gene variants associated with human eye diseases.

## 3. Classical Therapeutic Approaches to Restore the Vision

There are many ophthalmological processes such as replacing the lens after a cataract and fluid leakage control from retinal vessels which are already considered the principle guideline in clinical care. Lately, gene therapy was shown to be effective in enhancing visual performance for individuals that suffer from LCA and other hereditary disorders [45]. Although there is huge progress in this field, from the patient’s perspective, this progress is very slow. This is due to adverse challenges that exist in the retina such as how complex it is and that there is no regeneration potential. Therefore, therapeutic procedures that are being used until now are very limited to and targeted only to impede the degeneration of vision. There are many reports each year about new therapeutic options and different procedures to treat IRDs that comprise stalling cell degeneration such as neuroprotection, spurring the remaining retina-like retinal prostheses, and supplying the cells with a duplicate that has the same function as the gene of interest, e.g., gene therapy. Retinal prostheses are a type of bionic eye or implant electronic device that stimulate the sensation of vision. These implants have been used to treat retinitis pigmentosa. The stimulation of visual sensation can be achieved in two ways, either directly to ganglion cells or by using electric impulses that excite the internal retina, which is directed to the ganglion cells, requiring a thorough analysis of the retinal reorganization as it can vary from one patient to another. Normally, many factors constitute the retinal prostheses such as a small camera, an image processing unit, a stimulator chip, and an intraocular electrode array. The camera takes a photo of the field of sight and transmits it to the image processing unit, where the image is translated into the right stimulating pattern that is sent to the stimulator chip which delivers the stimulation to the intraocular electrode array [46]. Recently, FDA approved the second generation of external hardware (glasses and processing unit; Argus2s) to practice with Argus II implants though Second Sight has suspended Argus II implants since 2019 in favor of visual cortical prosthesis systems with the electrode neural interface moved from the retina to the visual cortex [47,48]. Prosthetic devices can be used to target other parts of the visual pathway; for instance, the optic nerve, lateral geniculate nucleus, and visual cortex have been developed that may be of specific assistance to patients with severe retinal damage. Examples of the ongoing studies with cortical prostheses are Orion Cortical Visual Prosthesis System (Second Sight; NCT03344848) and Intracortical Visual Prosthesis (NCT0463438) [49,50]. Choosing a therapeutic procedure for implementation relies on the stage of disease, however none of the therapeutic strategies that are being used have ever reported complete restoration of normal vision that was already lost [51]. For example, when retinal degeneration reaches the late stage, the retina will contain only a few photoreceptor cells which prevent the use of treatment options that depends on the existence of healthy cells, so when the degeneration reaches an advanced stage, the only proper therapeutic options are cell transplantation, retinal prostheses, and optogenetics [52]. Neuroprotective therapeutic strategies have been used to enhance the survivability of neuron cells by inhibiting apoptosis directly or by strengthening the means that enhance cell survival [53]. The basic issues were associated with the low ability to copy the results of the clinical trials which are usually done on animal models that have distinct compositions than humans [54]. Optogenetics is a method used to insert light-sensitive proteins into cells that facilitate neural activity control and can be monitored by using light. These light-sensitive proteins encoded by opsin genes, especially microbial opsins can serve as light-responsive ion pumps or sensory receptors [49,55]. This method can be used in the recovery of sight which is due to the degeneration of photoreceptor cells, however the leftover cells from the bipolar and ganglion retinal cells that existed in the retina can be targeted by optogenetics [51]. Moreover, optogenetics obligates the use of goggles during treatment since it is used for projecting the visual field onto the retina in intensity and spectrum that coincides with the opsins used; however, opsins are limited to a narrow range of light intensities [56]. The stimulation of retinal cells in invertebrates can occur by injuries and it is possible to regenerate the photoreceptor cells. Typically, adult mammals have stem cells that are present in the retina but they cannot spontaneously re-enter the cell cycle [57]. In addition, injuries can cause cell degeneration instead of cell stimulation. So, the transplantation of retinal progenitor cells or stem cells to the damaged retina has a large prospect for the treatment of retinal diseases [58]. Finally, stem cell transplantation is an interesting prospect for treating late-stage retinal degeneration but the level of organization of the retina cells must be analyzed carefully before the commencement of treatment since a high level of organization can hinder the development of new cells that can form networks inside the retina [59]. However, there are still many unknown factors in this strategy such as the way to increase cell survival, integration, and differentiation in a patient’s retina [60]. The classical therapeutic approaches to restore vision are time consuming, laborious, less efficient, and costly. These limitations can be overcome by an efficient and cost-effective RNA guided gene-editing technology (CRISPR/Cas9) that is currently in progress to overcome the limitations of conventional recombination methods [3].

## 4. CRISPR/Cas9 Is a Promising Strategy to Restore the Blindness

### 4.1. CRISPR-Cas System; Development, Components, and Mechanism

In 1987, scientists have observed a rare structure comprising 29 vastly conserved nucleotides located in the 3′-end adjoining region of the iap gene in *E. coli*. Afterward analogous repeats have been found in many strains of bacteria and archaea by random sequencing of their complete genomes [61]. These sequences of a clustered repeat were termed short regular spaced repeats (SRSRs) that are frequently spaced via unique intervening sequences of constant length and have been classified as an exclusive gene family existing in the immune systems of prokaryotes providing them an acquired resistance against viruses [62]. The existence of these inward short, inverted repeats in the repetitive units shows the same physiognomies of known sites intended for specific DNA-binding proteins. Many queries have been raised from these findings, such as whether their existence may be due to some of the earliest sequences or have been diverged through the process of evolution.

At the beginning of the 20th century, this innovative repetitive family ‘CRISPR’ was discovered via in silico analysis by revealing many genes that existed in line with the cluster repeats that were denoted as CRISPR-linked genes or Cas genes. Later on, these gene families of CRISPR were recognized as stably maintained secondary structures of RNA demonstrating efficient preservation through the process of evolution [63]. These structures are considered to perform significant functions in the prokaryotic gene repair and regulation to instigate their defense against different viruses. Subsequently, tremendous applications of these structures have been studied to edit animal and human genes using stem cells in order to generate restrictive mutations to find out the inside working environment of the cell. Certainly, the applications of the CRISPR-Cas9 system seem to be beyond our imagination. 

The CRISPR-Cas9 system was first investigated in prokaryotes which are using it as an adaptive immune system to counter phage viruses. The system comprises two important constituents, e.g., an endonuclease (Cas9) and a sole guide RNA (sgRNA) used to detect a specific sequence in the genome. The Cas9 act as an endonuclease by cleaving a specific sequence in the genome containing two catalytical positions (RuvC and HNH), nuclease domains that produce DSBs by cleaving the reverse strands of DNA [51]. SgRNA contains two components to enhance its action that comprises CRISPR RNA (crRNA) and an intergenic trans-activating crRNA (tracrRNA) used as a guided tool required for the activity of Cas9 activity. The crRNA contains a sequence ~20 bp that binds to its target sequence following the complementary base pairing rules of Watson–Crick model. The crRNA recognizes the target based on a DNA sequence that is short and conserved called protospacer-adjacent motif (PAM) which is positioned adjacent to the target DNA. SgRNA can be simply designed and cast off rapidly. It unites the Cas9 as a ribonucleoprotein (RNP) complex in the CRISPR structure and SgRNA directs the Cas9 to the targeted position in the anticipated gene and at that point, Cas9 exactly cut the DNA at 2–3 nucleotides earlier than the PAM sequence to generate DSBs. Generally, HDR or NHEJ are the two key mechanisms in DNA repair therefore DSBs can be repaired by any of these mechanisms. Similarly, the sgRNA may be restructured by modest variations in the DNA sequence to retarget novel DNA sequences. Furthermore, the CRISPR-Cas9 system can be used to change numerous genes through a solitary nuclease employing diverse sgRNAs simultaneously (Figure 1). 

An alternative form of Cas 9 is Cas9 nickase (nCas9) which varies in a single point mutation at position D10A or H840A as of Cas9 and cuts single-stranded DNA sequences targeted by sgRNA. Likewise, the double mutation in H840A and D10A of the HNH and RuvC domains of Cas9 may disable the endonuclease action of Cas9 and can reinforce dead Cas9 (dCas9) [42] which are unable to cut the DNA positions in order to generate DSBs; Therefore, it may be attached to transcriptional activators to make a CRISPR activation (CRISPRa) system capable of altering the gene expression by rewriting the epigenetic marks on the target gene; moreover, It may be attached with transcriptional inhibitors to develop a CRISPR inhibition (CRISPRi) system used to overwhelm the process of transcription [64].

### 4.2. CRISPR-Cas System Components Delivery to Ocular Tissues

Different types of injection routes have been used to deliver to ocular cells such as topical administration, intracameral injection, and subconjunctival, that are commonly used for gene transfer, and that can be used for the transfer of CRISPR/Cas9 components. The plasmids DNA gene has been delivered for transgene expression through topical administration. The intracameral injection is usually used to transfer the genes to the interior part of ocular cells such as the Conroy endothelial, and ciliary body. The intravitreal rout injections are used to transfer the genes to the inner retina cells, especially into the ganglion cells. The subretinal injections have been used successfully to transfer transgenes to the outer parts of the retina. Multi variants types of vectors have been designed and used to transfer the Cas9 and RNA components to targeted ocular cells. The Cas9 as of *Streptococcus pyogenes* (SpCas9) is considered as the most studied ortholog however, due to its greater size (4.2 kb), confining its capacity for in vivo use to be packed with gRNAs into a single adeno-associated virus (AAV) vector. To overcome this constraint, packaging into lentiviral vectors can be employed for twin AAV vectors to express the whole SpCas9 distinctly as of gRNAs, or to use the split-Cas9 by separating the SpCas9 expression on its messy element V713–D718 and remaking the entire protein via trans-splicing of split-intein protein [65]. Furthermore, SaCas9 of *Staphylococcus aureus* and CjCas9 of *Campylobacter jejuni* contain minor Cas9 orthologs that can be combined and accompanied by gRNAs simultaneously in the AAV vector [66]. The first time the CRISPR technology was used in the human eye was in the year 2019. The clinical test has assessed the subretinal AAV-facilitated transport of SaCas9 combined with a couple of gRNAs to mark a profound intronic mutation in the CEP290 gene in order to treat type 10 Leber congenital amaurosis (NCT03872479) [67]. Remarkably, even with the theoretical gain to use solitary viral vector for clinical translation, different researchers have observed that genome editing efficacy of the smaller Cas9 alternatives is mediocre with regards to the dual-vector delivery of SpCas9 and gRNAs [66].

Even though viral vectors can be efficiently used as genome editing tools in the retinal cells, the constant viral expression of Cas9 endonuclease may cause off-target effects. In contrast to conservative gene expression or biofactory approaches, the CRISPR systems no longer entail enduring transgene expression, wherever the continued existence of Cas9 may activate indifferent mutations. A different approach intended for clinical use is the straight delivery of recombinant Cas9 proteins along with gRNAs as ribonucleoprotein complexes (RNPs) in the eye that can instantly influence the cleavage of DNA to be quickly degraded in cells, thereby minimizing the off-target effects as well as cytotoxicity [68]. The straight use of RNPs in human cells proved to be an effective gene cleavage method compared to the transfection of plasmid by means of ~79% on-target mutation with less off-target effects and if carried out in the subretinal region for targeting *VEGF* in the mouse eyes, it resulted in ~40% decline in a CNV laser-induced model. Regardless of the RNP advantages, the Cas9 protein delivery into the nucleus is still a difficult task, mostly attributable to endosomal entrapment in the cytoplasmic matrix. In this regard, the cell-penetrating peptides (CPPs) can enhance the transport efficacy by ~80% [69]. In summary, due to size limitation capacity, mostly the adenosine-associated viral vector (AAV) has been widely used to transfer Cas9 and gRNA cargo into ocular cells. The AVV has a better balance of efficiency and stability. The non-viral vectors can be used in some circumstances, such as off-target effects and immunogenic reactions. With the rapid development of ocular gene delivery vectors, the advanced tailer-designed vectors will enable more efficient and effective CRISPR/Cas9 delivery. 

#### 4.2.1. CRISPR to Treat AMD

AMD is a multi-genetic ailment and the key source of blindness worldwide. A neovascular form of AMD is the wet AMD caused via irregular growth of choroidal vessels in the macula area of the retina ensuing damage to central vision. The macula is controlling the color vision and bright light actions due to their abundant cone photoreceptors [45]. The excess proliferation of vascular endothelial growth factor (*VEGF*) causes neovascularization in AMD, therefore anti-VEGF mediators turn out to be an excellent therapy [46]. Recently such types of mediators (aflibercept, bevacizumab, and ranibizumab) can be given as an intravitreal injection to treat wet AMD patients [47]. AMD patients can be treated with the AAV-CRISPR tool designed based on CjCas9 (*Campylobacter jejuni*) [38,70,71] as well as type-V CRISPR-Cas systems, LbCpcf1 nucleases. The CjCas9 gene with its matching sgRNA sequence and marker gene has been packaged into an AAV vector. CjCas9 delivered through AAV can specifically cut a restricted number of sites in the human or mouse genome that can cause desired mutations in the RPE cells. In this regard, the Vegfa or Hif1a gene in RPE cells can be targeted by CjCas9 to reduce the size of laser-induced choroidal neovascularization and this approach can be developed into in vivo genome editing therapy for AMD. After six weeks an AAV-CjCas9 injection intravitreally has resulted from an Indel efficiency of 22 ± 3% in Vegfa and 31 ± 2% in Hifla genes, respectively. Additionally, the outcomes of the Indels have been observed at the protein level e.g., substantial decline in VEGF-A protein was detected in RPE cells as compared to the control set. Another research group used lipofectamine 2000 for the delivery of sgRNA/Cas9 expressing plasmid with Cas9 RNPs which showed 82 ± 5% and 57 ± 3% of indel in NIH3T3 and ARPE-19 cells, respectively. In this study, Cas9 RNPs have been observed to be highly active w.r.t plasmid on the second day of transfection. After treatment with Cas9 RNPs, a reduction in the VEGF A mRNA and protein levels of 40 ± 8% and 52 ± 9% has been observed in the adult retinal pigment epithelial cells (ARPE). To evaluate the in vivo efficacy, Cy3-characterized RNPs have been delivered through intravitreal injection. After three days of injection, Cy3 dye was accumulated into RPE cells with detection of 25 ± 3% of indel at the delivery site in RPE cells. Furthermore, to mimic wAMD a CNV mouse model has been developed using laser tracked by an injection of Cas9 RNPs in the subretinal region. Three days later 22 ± 5% indel was detected in RPE cells for the *VEGFA* gene. The treatment with Cas9 RNPs has tremendously condensed the CNV area by 8 ± 4% along with a reduced level of *VEGF A* protein (Figure 2) [72].

Recently a lentiviral system has been reported to deliver mRNA encoding an extended Cas9 protein (SpCas9) and gRNA instantaneously named mLP-CRISPR. The mLP specifies mRNA-carrying lentiviral particles that are used to prevent the progress of wAMD in a laser-persuaded CNV mouse model. Subretinal injection of mLP-CRISPR displays an enhanced tissue specificity in the retinal pigment epithelium (RPE) cells that are considered to be the main cause of *VEGFA* in the outer portion of adult eyes. Additionally, mLP-CRISPR did not induce anti-Cas9 IgG in blood or T-cell permeation in the eyes. An injection of mLP-CRISPR has disrupted 44% of Vegfa genes in RPE while decreasing 63% of the laser-induced CNV area in the wAMD mouse model. This has been done by using a gene-editing tool without any off-target effects. This type of mLP technique can be used to carry mRNA encoding numerous Cas9 nucleases, base, prime, and epigenome editors [73]. 

#### 4.2.2. CRISPR to Treat the Glaucoma

Glaucoma is an assemblage of eye diseases caused due to advanced and permanent disintegration of the ganglion cells in the retina which axonal projections establish the optic nerve [74]. At present, it is considered the foremost source of irreversible blindness globally [71] that could affect >76 million people by the year 2020. Presently, glaucoma is mainly treated in clinics by reducing the intraocular pressure (IOP) in affected individuals by medicine, laser action, or surgical treatment. However, surgical and laser therapy brings risks that frequently need additional interference or combinational tactics supplemented with more topical therapies during a patient’s lifetime [47]. The patients frequently require consistent treatment through numerous types of eye drops that can be used many times daily. It means that patient treatment becomes a challenge with the passage of time and even numerous of them have experienced continuous visual loss instead of their reduced IOP [48,49]. Consequently, novel therapeutic methods can be developed that can be obtainable by an injection into the eye directly with enduring or perpetual beneficial consequences. A considerable number of adult individuals with glaucoma have an indistinct, varied cause of disease linking numerous genetic, environmental, and individual risk factors [38,75]. Due to these reasons, the commencement of glaucoma in adult individuals can be controlled by gene therapies that are mainly focused on neuroprotection, which encompasses the reduced loss of RGCs by changing their physiological status in order to reduce the severity of the disease. This can be achieved by either enhancing the action of innate survival pathways in RGC or inhibiting the development of programmed cell death.

With the recent developments in specificity and efficacy of CRISPR-mediated genome editing technology, there is a hope to target these mutations. Currently, researchers have targeted the dominant *MYOC* mutations in a mouse model of myocilin-linked POAG through adenoviruses that are expressing the CRISPR/Cas9 components (*Ad5-cas9* and *Ad5-crMYOC*). In this study, Cas9 has successfully knock-out the mutant *MYOC* gene, reduced IOP in the eyes of treated mice, and stopped further glaucomatous damage in mouse eyes. Similarly, the same constructs have been used for the treatment of trabecular meshwork tissue in the human eyes cultured ex vivo. A decrease in the myocilin mRNA suggests the possibility of using this technology to overcome the problems of patients with *MYOC* mutations [76].

Primary open-angle glaucoma (POAG) has been considered the main global source of blindness, comprising an imperative risk factor of elevated intraocular pressure (IOP). The pathological variations in the trabecular meshwork (TM) increase POAG IOP linked with changes caused by an enhanced level of *TGFβ2*. Recently, CRISPR interference has been used to explicitly deacetylate histones in order to decline *TGFβ2* in the TM. The CRISPR interference system has been observed to constrain TGFβ2 expression in human TM cells through accurately designing sgRNA that have targeted the *TGFβ2* gene promoter. The sgRNA has targeted the CMV promoter of the Ad5-CMV-TGFβ2 viral vector and it has been observed that lentivirus-mediated KRAB-dCAS9 and sgRNA expression were capable to constrain Ad5-CMV-TGFβ2-induced OHT in the eyes of C57BL/6J mice (male and female). The reduction in OHT was linked with a diminished level of *TGFβ2* and extracellular matrix proteins in the mouse eye. These findings suggest that CRISPR interference can be used as a tool for gene inhibition containing the therapeutic potential for the treatment of TGFβ2-induced OHT [77].

Currently, a practical gene therapy tool has been used to reduce IOP via specific disruption of the aqueous humor synthesis in the ciliary body ensuing an intravitreal injection. In this regard, explicit gene editing was done by means of the smaller *S. aureus*-derived CRISPR-Cas9 that is proficient to be carried in a single recombinant AAV vector which is considered a benchmark by US-FDA and has been approved for visual gene therapy. In order to target a gene vital for a well-maintained functional development relatively to correct a single explicit mutation, a collective approach deprived of excessive personalized tactic is manageable. IOP decrease can be accomplished through knock outing Aquaporin 1 (*Aqp1*) in the ciliary body. Aquaporins are a group of water-transferring transmembrane proteins that are broadly expressed all over the human body and transgenic mice lacking *Aqp1* have shown lesser IOP due to abridged influx to form aqueous humor deprived of contrary consequences [78].

#### 4.2.3. CRISPR to Treat Retinitis Pigmentosa (RP)

RP is considered an important source of progressive blindness affecting 1 in 4000 individuals [50]. The rod-cone dystrophy is a usual RP that is recognized via shaft vision ultimately leading to the loss of peripheral vision. Its early symptoms contain nyctalopia which leads to night blindness and patients have complications in adjusting to the dark that happened to the injury of rod function in early childhood [51]. Due to the damaged photoreceptors, RPE begins to lose its pigment eventually leas to the buildup of intraretinal melanin deposits, which appeared as a bone spicule conformation. Though, the central vision is still integral till the last stages. Different genetic modes are responsible for its transmission, such as autosomal dominant and recessive or X-linked and are heterogeneously linked with mutations as a minimum of 79 genes [79]. There are primarily two types of RP which are MERTK connected and RPGR X-linked. The apical membrane of RPE comprises photoreceptors that are sensitive to light and mediated by MERTK (Mer tyrosine kinase) receptors that are engaged in the rods and cones phagocytosis. To play an effective role, the continuous recycling of these photoreceptors is very important that is interrupted by mutations in MERTK, which proceed to degradation and ultimately decline of the photoreceptors [41]. It has been observed in meta-analyses that ~3% of MERTK type RP are a result of autosomal recessive transmission which are causing macular atrophy and childhood photoreceptor abnormality [43,44]. Alterations in the RP GTPase regulator (*RPGR*) that is an X-linked RP (*XLRP*) has been observed in 1 out of 3500 individuals. RPGR, as well as the δ subunit of rod cGMP phosphodiesterase, controls the proteins and its disorder is causing continuous damage to the central vision and progressing to night blindness [51]. 

CRISPR/Cas9 technology has been used in some in vitro and in vivo studies to treat RP e.g., applied in a rat model of adRP to remove mutation in the rhodopsin gene (*RhoS334*). In this experiment the sgRNA/Cas9 plasmid has been used to target exon 1 closely upstream of a PAM exclusive toward the *RhoS334* locus was directed intravitreally in S334ter-3 rats. In two different rats, a cleavage efficacy of 33 and 36% was confirmed in transfected retinal cells through genome analysis. Subsequent injection of sgRNA/Cas9 plasmid has confirmed an enhanced visual insight and widespread protection of the retina through immunohistological examination [80]. Moreover, the same strategy was used to edit the RHO gene mutations. In this research, an intended plasmid containing an insert for 2 sgRNAs targeting the RHO gene was used to generate DSB and subsequently NHEJ. As a result of this research, the RHO gene was successfully edited with additional downregulation of the expressing RHO protein. In another study, CRISPR/Cas9 system was used in iPSCs attained from a patient with photoreceptor degeneration to treat XLRP to correct *RPGR* (a pathogenic point mutation). In this research, 21 different sgRNAs have been screened for editing, and among them, g58 was observed as utmost activity. Thus, the plasmid containing g58/Cas9 was considered to transfect the iPSCs together with *RPGR* single-stranded oligo deoxy ribonucleotide (ssODN) that plays a supportive role in the HDR pathway. After deep sequencing analysis, the data showed an effective modification of mutation in 13% of the transfected cells [81]. Additionally, the premature stop codon TAG was successfully substituted by the wild-type codon GAG encoding glutamate at position 1024. In contrast, the untransfected iPSCs did not show any variations in the mentioned mutation. It was determined that the 13% correction rate was meaningfully productive that can be further enhanced by reducing error-prone NHEJ through DNA ligase IV inhibition on the DNA cleavage site. Moreover, a CRISPR/Cas-based approach was established to edit RHO gene mutations in the ADRP mouse model using a plasmid comprising an insert designed for two sgRNAs to target the RHO gene (exon 1) containing P23H dominant mutation. Primarily, HeLa cells have been used for gene editing in vitro showing an indel frequency of 70%, 76%, and 82% by means of sgRNA1, sgRNA3, and 2sgRNA, respectively. Further, the Real-time Taqman PCR has been used to observe RHO expression where 35%, 25%, and 20% of reduced expression level has been detected in the cells treated with sgRNA1, sgRNA3, and 2sgRNA, respectively. Later on, the CRISPR/Cas plasmid has been used by electroporation in P23 RHO transgenic mice comprising 2sgRNA together with green fluorescence protein (GFP) to achieve subretinal expression. To evaluate Cas9 expression, the GFP expressing section of the retina has been isolated where the expression was inadequate in the cells articulating GFP together with 84 edited sequences [81]. 

The scope of treatment is the inadequate beginning embryonic day (E) E16 to P2 due to the rate of retinal deterioration in adRP patients that is heterogeneous. Therefore, an ablation therapy for long duration could be useful to maintain vision in animal models. Thereby, a slower degenerating adRP model having a common adRP mutation (rhodopsin *P23H*) has been used for clinical application. In adRP patients, the *P23H* (*RHOP23H*) is a very common mutation that can be observed in the rod cell-specific gene rhodopsin (RHO) which is an effective mark to study mutation-specific ablation approaches via CRISPR [10,13]. Usually, an excessive level of rhodopsin misfolding (class II mutation) and mistrafficking (class I mutation) as a result of *P23H* mutation is accumulated in the endoplasmic reticulum (ER) [14,15,16]. The *P23H* line-3 rats experience photoreceptor degeneration from P15 which shows the level of vision retained with the passage of time equal to decades in patients accepted for treatment at various disease stages. In this research, the subretinal delivery of AAV-CRISPR/Cas9 has been shown as a benign approach that can cure photoreceptors and vision intended for 15 months in Rhodopsin *P23H-3* rats (Figure 3) [82]. 

#### 4.2.4. CRISPR to Treat Leber Congenital Amaurosis (LCA10)

LCA is considered a high overwhelming retinal dystrophy due to its potential of causing congenital blindness in <1 year of age. Until now 14 mutant genes have been known based on linkage examination, homozygosity mapping, and genome scrutiny in LCA patients among which 70% are children with retinal degeneration [40]. LCA patients are generally connected with plain imperfections comprising rambling eye movements termed nystagmus. In children, the symptoms are usually sluggish pupil reactions and deficiency of electroretinographic reactions [41,42]. Genes intricate LCA encode the proteins that are accountable for retinal roles comprising vitamin A cycling (*LRAT*, *RPE65*, *RDH12*), photoreceptor morphogenesis (*CRB1*, *CRX*), guanine synthesis (*IMPDH1*), phototransduction (*AIPL1*, *GUCY2D*), outer segment phagocytosis (*MERTK*), and intra-photoreceptor ciliary transport progressions (*CEP290, RPGRIP1, LCA5, TULP1*) [40]. Until now, the investigated genes for LCA stand mutations in the RPE (*RPE65*) gene that is encoding retinoid isomerase [30], although, the maximum going on mutations are connected with the *CEP290* (15%), *GUCY2D* (12%), and *CRB1* (10%), respectively. About 20% of north-western Europe patients have an intronic *CEP290* mutation (*p.Cys998X*). In the mice, an AVV-CRISPR system has been studied to investigate the in vivo action of autosomal dominant retinitis pigmentosa (adRP) and LCA10. In this research, the AAV-SpCas9 vector was carried through subretinal injections targeting the rhodopsin (RHO) or *CEP290* and *Nrl* (neural retina leucine zipper transcription factor) gene using a mouse model for adRP. The results are very promising which show an enhanced level of spCas9 protein expression in the retinal cells of mice for a period of nine-and-a-half months. Additionally, the researchers have arrayed diverse AAV serotypes besides changed vector doses and have found a real repair of RP or LCA10 phenotype lacking off-target properties or adverse toxic responses [43,44]. The same approach has been used successfully in human cells to resolve the RHO gene mutation. This study confirms CRISPR/Cas9 as an effective system to target gene/alleles in a well-organized way indicating that it may be cast off for the treatment of RP and further dominant human genetic disorders [83]. 

#### 4.2.5. CRISPR to Treat the Usher Syndrome

Usher syndrome is considered a frequent form of syndromic IRD with a prevalence of 1 out of 20,000 individuals with its exclusive characteristics comprising RP and hearing damage [70]. Based on the development and intensity of the hearing injury along with the RP outset, this heterogenous syndrome is segregated into three subtypes which are usher syndrome type 1 (*USH1*) utmost critical, *USH2* frequently observed with modest to severe symptoms, and *USH3* considered as a moderate phenotype that varies case by case regarding disease outset and its progression [71]. *USH1* is considered to be the main source of deaf-blindness that is inherited in an autosomal recessive manner in humans causing vestibular ailment, deep congenital heart loss, and RP. *USH1* is triggered as a result of mutations in myosin VIIA encoding an organelle transport protein in the RPE [46]. CRISPR/Cas9 gene editing was proposed as a gene-editing tool to reinstate the *c.2299delG* mutation in the *USH2A* gene. The gene-editing experiment was carried out in human dermal fibroblasts (*HDFs*) cells taken from a *USH2* patient having *c.2299delG* mutation. By means of nucleofection, a Cas9 RNPs containing 15 µg Cas9 and 20 µg sgRNA has been transfected into the normal individual *HDFs* producing 18% indel rates. Afterward, delivery of RNPs combined with ssODN-2299 has produced 5% HDR efficacy. Correspondingly, patient HDFs with *c.2299delG* mutation have been transfected with ssODN having a WT sequence with a deleted PAM sequence resulting in 6% indel efficacy with 2.5% deletion of HDR [72].

Recently a pig model of *USH* imitates the mix of deafness, vestibular dysfunction, and vision loss found in *USH* individuals. Visual impairment has been observed in *USH1C* pigs through ophthalmologic checkups, studying their behavioral examinations go together with morphologic changes in the photoreceptor cells. Additionally, the photoreceptor and primary cilia of fibroblasts have been constantly extended in *USH1C* pigs which could help to conduct gene-editing experiments as potential therapeutics. Investigative gene therapeutic tools can verify the improvement of visual function in *USH1C* pigs (Figure 4) [84].

## 5. Base and Prime Editing: Advanced Gene Editing Tools to Cure Blindness

In the last decade, a lot of safety concerns have been reported in gene-editing studies regarding off-target effects due to double-strand break (DSB) in CRISPR/Cas9 experiments. The DSB generates DNA lesions which leads to deleterious chromosomal rearrangements, duplication, inversion, deletions, and translocation that generates instability in the edited genomes [85]. However, the Cas9-nickases as Cas9 variant that generates the single strand break was the first preference of researchers for the integration of foreign DNA pieces through HDR. Before the development of CRISPR/Cas9-based gene-editing genetic therapies, there would be deep knowledge of DNA repairs system such as NHEJ. The NHEJ mostly leads to off-target effects, which is why further development of highly sensitive unbiased gene-editing techniques is a dire need of the current research. As an alternative, CRISPR/Cas9, which should edit the genes without the DSB and NHEJ, was investigated. To reduce the off-target effects, recently researchers have developed new precise gene-editing technologies, such as base and prime editing. The working principle of the base editor consists of fusing the Cas9 nickase with the cytosine or adenine deaminase which converts the base pair such as C to A or A to G without generating the double-strand break in the DNA strands. The first base editor was developed by combining the Cas9-nickase with the cytosine deaminase enzyme, APOBEC1 which converts the C to T. Through a direct evolution process the RNA adenosine deaminase was generated and the second base editor was developed by combining the Cas9 nickase with RNA adenosine that converts the A to G. Recently a dual base editor has been generated which can covert adenine and cytosine basis simultaneously on DNA strands. As compared to CRISPR/Cas9 tool, which only work in dividing cells, conversely the base editor also works in non-dividing cells. The base editors only cause nucleotide conversion at a specific point rather than DSB by Cas9, and there is a negligible chance of insertions, deletions, and transversion chromosomal rearrangement in base-editing experiments. Recently a new technology, prime editing has been developed by combining the Cas9-nickase and engineered reverse transcriptase. The Cas-nickase is guided through prime editing guide RNA (pegRNA) consisting of 3′ end-extended gRNA. The 3′end extended portion of pegRNA act as a primer for the functioning of the reverse transcriptase (RT) initiation. After finding the complementary sequences the RT activated that synthesized the complementary DNA (cDNA), which is incorporated at target sites after a series of flap removal events. Prime editing is a versatile technique that can introduce small deletion and insertion efficiently. The highest efficiencies with no off-target effects were observed in editing the HK293T cells. In the last two years, base and prime editing have been employed for eye diseases [86,87,88]. 

Lebar congenital amaurosis (LCA) disease characterized by the loss of function mutation in the *Rpe65* gene, which is the crucial factor in LCA emergence. This non-sense mutation was corrected through the base editors efficiently with zero off-target effects in mice that helped in the regeneration of visual chromophores which profoundly enhanced and restored the visual function in mice. The adenine and cytosine base editors were delivered and transduced efficiently into the retinal cells. However, small insertion and deletions were observed in the mice [89]. The prime editing has successfully edited the Dnmt1 gene in mouse retina with fewer off-target effects. Recently, six autosomal inherited genes associated with inherited retinal diseases (IRD) have been reported [90]. Primer editing has rectified 89% of genetic mutations that could be employed to modulate six pathogenic variants of IRD. As the new enzymatic modifications are coming day by day to reduce off-target effects, the base and prime editors would be at the forefront of the development of regenerative therapies.

## 6. Limitations of CRISPR/Cas9 in Clinical Applications to Cure the Blindness

Currently CRISPR/Cas9 has shown better potential to cure blindness. However, it poses inherent issues of controllability and stability that needs to be resolved before its clinical application [91]. The technology specifically targets a genome with high editing efficiency, facilitating quick and easy gene targeting and screening of recombinant viruses. The DNA repair induced by CRISPR-Cas9 seems to be quick, effective, and controllable, with recombination stability and fidelity in the case of multisite editing [92]. 

### 6.1. Delivery

The delivery of CRISPR/Cas9 machinery to the target cells is influenced by safety and therapeutics efficacy. The viral and non-viral strategies have been used to deliver the CRISPR/Cas9 components to targeted cells. In viral delivery, mostly three types of vectors, (1) adenoviral, (2) lentiviral, and (3) adenosine associated viral (AAV), have been used to deliver the genes of Cas9 and gRNA to a specific target locus. The adenoviral method has a harmful potential to trigger an immune response in host cells [93]. The lentiviral and AAV methods causes immunotoxicity, oncogenes insertion, and random mutations. Multiple types of mutations have been generated using viral vector deliveries such as deletion of Rep protein that has increases the transducer efficiency, but still the immunogenic problem sustained [94]. The AAV delivery is limited due to packaging capacity that makes AAV vector difficult to use for transferring Cas9/gRNA with selectable markers and regulatory elements such as single strand homologues DNA for gene knock-in in the retinal cells. Newer approaches have been explored to reduce the immunotoxicity, oncogenes insertion and module the packaging capacity of AAV [95]. The limitations associated with viral delivery methods such as immune system trigger could be compensated through the use of non-viral delivery methods such as nanoparticles. Numerous nanoparticles-based delivery approaches have been employed and approved by FDA for treatment of eye diseases which are under clinical trials. These include liposomes (NCT00121407) to treat AMD, aptamer-polymer nanoparticles (NCT00549055) to treat the wAMD, and lipid-based nanoparticles (NCT03093701) to treat the molecular edema. The nanoparticles-based delivery methods can be modulated for targeted delivery, sustained release of cargo, and increased exposure time, but the application of nanoparticles delivery is limited by biocompatibility and low in vivo delivery [96]. In recent years, researchers have modulated the nanoparticles for stimulus-based delivery to targeted cells for CRISPR/Cas9 based gene-editing, that could be used in future to treat retinal diseases [97]. 

### 6.2. Specificity or Precision Editing

The specific or precision editing is also a limitation of CRISPR/Cas9. The HDR pathway which is highly crucial for gene insertion or knock-in has very low efficiency as compared to NHEJ for gene knock-out. Currently the efficiency of HDR has been increased through inhibition of NHEJ pathway by employing the modulators of NHEJ suppression enzymes such as Ku [98] and DNA ligases IV [99]. Other strategies to increase the HDR pathway efficiency by adding the single strand DNA homologous to targeted regions has increased the 60% chances of correct and precise/specific gene editing in zebrafish and mouse models [100]. It has been proved that homologous arm of 48 bp upstream and downstream can significantly improve the precise gene insertion in zebrafish genome through CRISPR/Cas9 gene editing [101]. Moreover, cell cycle stages can play a key role in determining the DNA repair system HDR or NHEJ. The HDR DNA repair pathway is restricted to S and G2 phases. Currently the S phase has been arrested for longer time than usual to increase the HDR efficiency through the addition of aphidicolin [102]. Advances in genome editing has allowed the use of base editors with specific gene editing capability to modulate the DNA repair system. In this new gene editing system, inactive dead Cas9 (dCas9) fused with an enzyme deaminase can catalyze and convert the nucleotides without the requirement of DNA repair system (HDR, NHEJ). Consequently, highly specific adenine base editors have been developed that convert adenosine to guanine and cytosine base editors which convert cytosine to thymine with precise gene editing [87]. Recently prime editing technology has been developed which can edit the genome in a precise way without the double strand break. Researchers have combined the Ca9 with reverse transcriptase enzyme and prime editing gRNA (pegRNA) which contain the RNA template to specifically insert the desired sequence [88]. In future studies, base and prime editors can be optimized for precise/specific gene editing in the eye cells to restore the vision.

### 6.3. Off-Target Effects

One of the major issues with application of CRISPR/Cas9 for gene therapy is unintended mutations or off-target effects. Sometimes these off-target effects are observed above 50% [103]. Many strategies have been adopted to mitigate off-target effects at multiple levels such as delivery, Cas9 variants, and modulation of DNA repair pathways. One of the more effective strategies is the use of the Cas9 variant, Cas9 nickase (Cas9n), which generates the single strand break at one side of DNA with one guide RNA and a second cut at an opposite site of DNA through second gRNA. The application of Cas9 nickase has generated less off-target effects compared to wild type Cas9 [104]. In many gene editing studies, spCas9-HF1 has been used with any off-targets detected in mouse and zebrafish models, as compared with wild type spCas9 nuclease [105]. Other types of Cas variants have been developed includes evoCas9 and HifiCas9 by altering the amino acids in Rec3 domain, which is involved in the recognition of homologous DNA nucleotides. This mutation has desensitized the Rec3 domain of protein and ultimately reduced unintended mutations with high editing efficacy [106]. In another strategy, gRNA has been used by reduction in size at 5′, GC content, and optimizing the sequence. Many algorithmic based computational tools have been developed, such as CasOFFinder, and E-Crisp allowing researchers to generate the sequences with most optimized gRNA nucleotides. These tools have significantly helped to reduce the off-target effects [107,108].

## 7. Conclusions and Future Outlook

In the past 10 years, CRISPR/Cas system has emerged as a revolutionizing genome editing tool that has been used to treat many ophthalmic diseases affecting the lives of thousands of patients. CRISPR/Cas9-based clinical trials are in progress to treat LCA10 to mutate the CEP290 [109]. Highly efficient gene disruption through CRISPR/Cas9 makes it a promising technique to treat the dominant mutations in rhodopsin and retinitis pigmentosa. A large number of point mutations can be generated through the HDR pathway. Numerous mammalian and non-mammalian eye disease model organisms have been generated to make the CRISPR/Cas9 a promising therapeutics tool to cure blindness. Current ongoing research in ophthalmology has shed a light on the pathogenic mechanisms of non-genetic eye diseases. In the near future, the development of highly targeted editing CRISPR tools in ophthalmology will be at the forefront to treat genetic eye diseases to cure blindness. However, the CRISPR tool application is limited due to off-target effects, heterogeneity of diseases, and ethical concerns. The early CRISPR/Cas9 tool relied on double-strand DNA repair, NHEJ, or HDR. NHEJ and HDR always lead to INDELS formation and off-target effects. The off-target effects can mutate the cancer tumor-suppressing genes. Therefore, it is crucial to predict the off-target effects before starting the experimental application of CRISPR/cas9. The off-target effects are determined by the PAM site and gRNA homologous regions. Many bioinformatics tolls have been generated that can be employed to predict the off-target effects before their application. The off-target effects can be significantly reduced with improved Cas9 variants, SpCas9-HF1, limited expression of Cas9, and improved genome editing methods, such as base editing and prime editing. In near future, researchers are trying to engineer Cas9 variants by direct evolution that will be able to decrease the off-target effects significantly [85]. The heterogeneity of eye diseases remains a challenging factor for their treatment. Many diseases have hundreds of causative mutations [110]. Using the iPSC stem cells and editing with CRISPR/Cas9 and then differentiating it into photoreceptors could be a better strategy to avoid heterogeneity. This strategy is still in its initial trials, but successful findings of these trails would support the use of CRISPR/Cas9 technology from laboratory to clinics. Researchers have focused on making the CRISPR/Cas9 as an effective and safe therapeutic tool that could be used in clinical trials. Advances in CRISPR/Cas9 technology such as base and prime editing could boost the application of CRISPR/Cas9 to treat retinal diseases in an efficient way with less cost. One of the main limitations of CRISPR/Ca9 technology would be the manufacturing and scale up for in vivo editing. With less cost and precision gene editing abilities the CRISPR/Ca9 will be a promising technology to cure blindness in the near future.

## Figures and Tables

**Figure 1 ijms-23-11482-f001:**
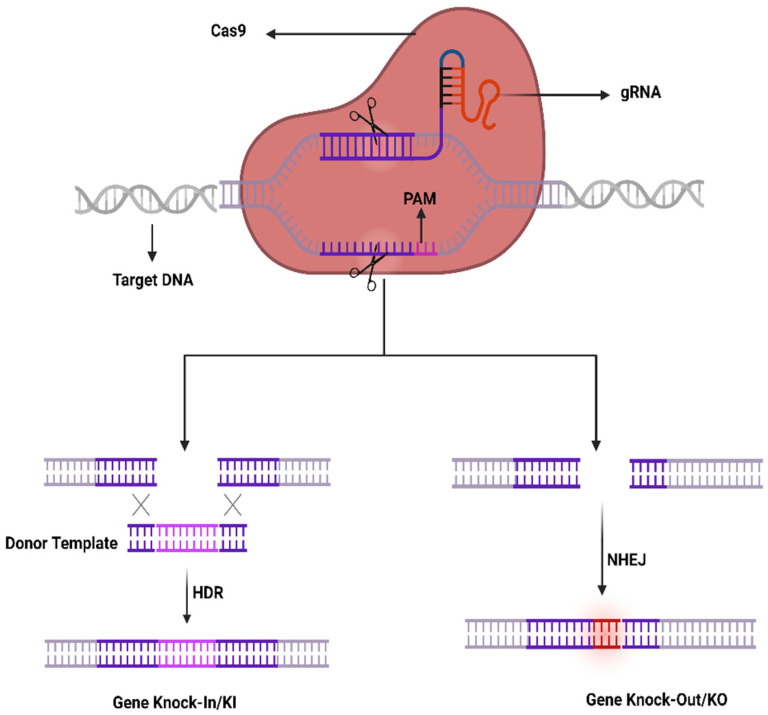
CRISPR/Cas9 working mechanism.

**Figure 2 ijms-23-11482-f002:**
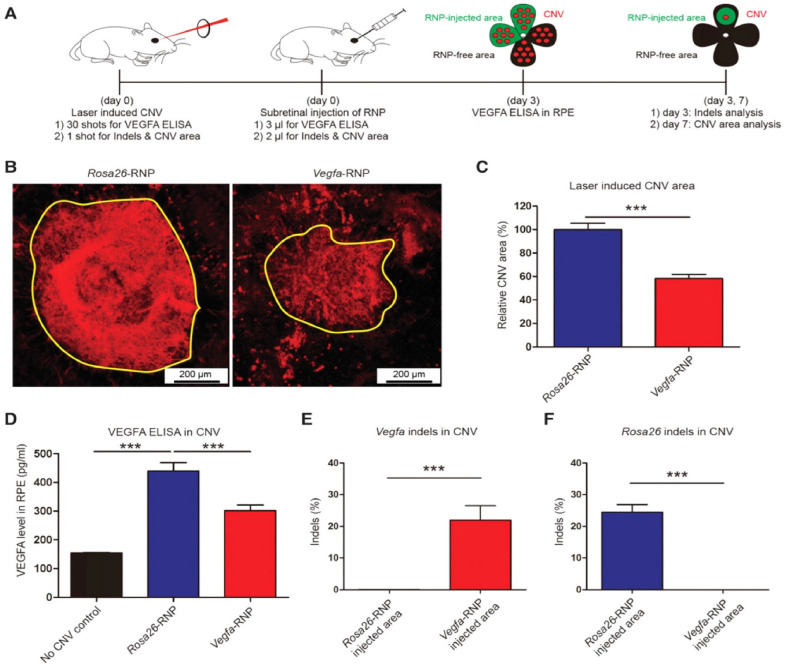
(**A**) Mice with laser-induced choroidal neovascularization (CNV) injected with Vegfa-specific Cas9 RNP analyzed at 3 days post-injection for Vegfa and 7 days post-injection for INDLES analysis. (**B**) The laser-induced CNV was stained with isolectin B4. (**C**) CVN area under Vegfa-RNA. (**D**) controlled and CRISPR/Cas9 edieted comparioson. (**E**) INDEL frequencies due to Vegfa-RNA injected. (**F**) INDEL frequancies at Rosa26 site. Student’s *t*-test: (***) *p* < 0.001 [72].

**Figure 3 ijms-23-11482-f003:**
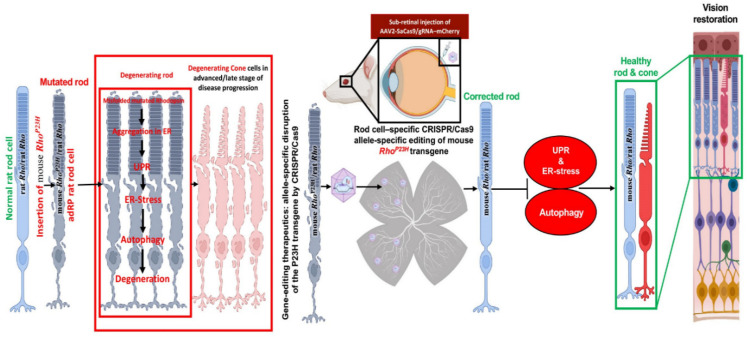
Schematic representation of CRISPR/Cas9 editing in the Rhodopsin P23H to restore and rescue long-term vision in mice [82].

**Figure 4 ijms-23-11482-f004:**
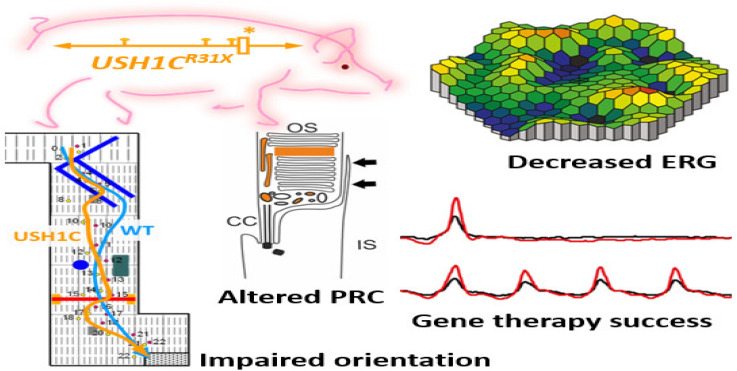
Schematic representation of disruption of photoreceptors architecture to rectify usher syndrome through CRISPR/Cas9 to restore the vision in pig. * Denotes the R31X disruptive mutation at USH gene in the pig model. ERG recordings in USH1C mutant pigs vs. WT demonstrate a significant reduction in the scotopic standard flash responses, dark is dim light and red with bright light stimulus. Matrix plot denotes the impaired orientation of mutant and wild type pigs [84].

**Table 1 ijms-23-11482-t001:** Gene variants associated with common human eye diseases.

No.	Eye Disease (Onset Age)	Gene Variants
1	AMD (50–60 years)	Nitric oxide synthase 2A *(NOS2A*), tissue inhibitor of matrix metalloproteinase 3 (*TIMP-3*), matrix metalloproteinase-9 (*MMP-9*), high-temperature requirement factor A serine peptidase 1 (*HTRA1*), excision repair cross-complementing group 6 (*ERCC6*), etc.
2	Glaucoma (>40 excluding congenital form in infants)	Mouse myocilin (*MYOC*), paired-like homeodomain2 (*PITX2*), paired box protein (*PAX6*), cytochrome p450-1B1 (*CYP1B1*), latent TGF-β binding protein-2 (*LTBP2*), etc.
3	Cataract (50–60 years)	Transmembrane anterior posterior transformation 1 (*TAPT1*), gem nuclear organelle associated protein 4 (*GEMIN4*), AP endonuclease 1 (*APE1*), major intrinsic protein (*MIP*), etc.
4	Myopia (progresses in ~ age 20)	Hepatocyte growth factor (*HGF*), mesenchymal-epithelial transition factor (*C-MET*), uromodulin like 1 (*UMODL1*), paired box protein (*PAX6*), etc.
5	Stargardt’s disease (early childhood to middle age)	ATP binding cassette subfamily A member 4 (*ABCA4*), crumbs homology 1 (*CRB1*), etc.
6	Retinitis pigmentosa (10–30 years)	Retinitis pigmentosa GTPase regulator (*RPGR*), precursor mRNA processing factor 3 (*PRPF3*), ATP/GTP binding like 5 (*AGBL5*), etc.
7	Marfan syndrome (newborn babies that may be later on)	fibrillin 1 (*FBN1*), transforming growth factor beta receptor 2 (*TGFBR2*), etc.
8	Polypoidal choroidal vasculopathies (55–65 years)	Complement 2 (*C2*), complement 3 (*C3*), complement factor B (*CFB*), erpin peptidase inhibitor clade G member 1 (*SERPING1*), pigment epithelium-derived factor (*PEDF*), etc.
9	Uveal melanoma (50–80 years)	Ubiquitin carboxyl-terminal hydrolase (*BAP1*), guanine nucleotide-binding protein G(q) (*GNAQ*), G protein subunit alpha 11 (*GNA11*), spliceosome factor 3B subunit 1 (*SF3B1*), eukaryotic translation initiation factor 1A X-chromosome (*EIF1AX*), etc.
10	Inherited optic neuropathies (X-linked appears in young males)	Complex I or *ND* genes, *OPA1*, *RPE65*, etc.

## Data Availability

Not applicable.

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
