# Peer review of "CRISPR/Cas9—A Promising Therapeutic Tool to Cure Blindness: Current Scenario and Future Prospects"

_ijms, 2022, doi:10.3390/ijms231911482_

Round 1

Reviewer 1 Report

Despite the interesting topic and the meritory work of the single Author, the help of an ophthalmologist may greatly enhance the content of this paper, which at times is messy and confused; in fact, it is clear that the Author does not know much about many of the ophthalmological topics that are reported in the manuscript (see for example pegaptanib discontinuation).

In the Introduction too much space is dedicated to ophthalmological diseases without arising much interest and poorly defining them, and the main topic of the paper (CRISPR/Cas9 System) is not clearly defined; Paragraph 3 briefly describes therapeutic approaches and should be improved, including the recent problems regarding Argus II retinal prosthesis.

Paragraph 4, the main point of interest of this paper, is very difficult to read and should undergo a major English revision, as it should be done in the whole paper.

Figure 1: replace “kncok” with “knock”.

Reviewer 2 Report

The review of current and future aspects of gene editing using CRISPR/Cas9 in the eye is very interesting and is beneficial to the readers to understand the perspective of genome editing in Ocular therapeutics.

1.While the the title of the review seems like more specific, I feel that author has generalized the topic by describing in detail all the retinal dystrophies which is not necessary for this review. Authors could have started with brief introduction and can go straight into the topic(As indicated in 4.1)

2. I would skip/ minimize all the info related to ocular genetic diseases and other therapeutic options as that is not the subject of the review.

3. Latest technologies like base and prime editing and challenges with the delivery of genome editing tools should be described in detail.

4.Methods of delivery should have different section and deserve elaborate mention.

5.Current status and perspectives of clinical trails using CRISPR genome editing approaches should be discussed,

Overall I feel that there is lot of information that need to be removed (Especially section 2 and 3) and specific information mentioned above should be added.

Minor:

Table 1 Describe onset of the disease with appropriate age ranges (Don't just say old people)

Round 2

Reviewer 1 Report

The Author has performed some revisions to the paper but the help of an ophthalmologist is needed and may greatly enhance the content of the manuscript; in fact, many ophthalmological topics that are reported in the manuscript still need to be better defined. The same approach should be performed in the paragraph regarding genetic mutations associated with ocular diseases.

Author Response

Comment: The Author has performed some revisions to the paper but the help of an ophthalmologist is needed and may greatly enhance the content of the manuscript; in fact, many ophthalmological topics that are reported in the manuscript still need to be better defined. The same approach should be performed in the paragraph regarding genetic mutations associated with ocular diseases.

Response: I am very thankful for the nice suggestion. The manuscript has been reviewed by an ophthalmologist and acknowledged accordingly in the revised manuscript.

Reviewer 2 Report

Authors have significantly made changes to the revised version of the manuscript which makes it more interesting to the readers. However, the review still suffers from lacking few important points such as 

1. Limitations of the CRISPR technology for clinical applications has to be mentioned which should include specificity of the technology, deliverability, non-specific genomic changes and how they can be addressed to overcome regulatory approvals for human use. 

2. In the same section how these limitations can be for some extent bypassed should be described. Such as use of modified CRISPR tools (reduction in their size), risk benefit ratio for off target effects.

3. Major grammar and spell check is still required especially for gene and disease names (for example Line 127 ABCA4 not ABCD4)

4. I think flow from one section from the other is completely missing, for example section 3 describes classical therapeutic approaches to restore vision while entering section 4 which describes use of CRISPR, section 3 should describe the limitations of classical therapeutic methods such as gene augmentation and why there is a need of alternative approaches.

5. what is the authors overall perspective of CRISPR genome editing in  future ophthalmological clinical application. Where do author see CRISPR technology over the coming decade in terms of clinical translation rate when compared to other approaches. This should be described in the conclusion part. I think future perspective is missing in the last section

Author Response

Comment 1: Limitations of the CRISPR technology for clinical applications has to be mentioned which should include specificity of the technology, deliverability, non-specific genomic changes and how they can be addressed to overcome regulatory approvals for human use.

Response: Thanks for your comment. The mentioned limitations and their promising solution have been addressed by adding another section 6 “Limitations of CRISPR/Cas9 in clinical applications to cure the blindness” from line number 646 to 730 (highlighted in green color) in the revised manuscript.

Comment 2:  In the same section how these limitations can be for some extent bypassed should be described. Such as use of modified CRISPR tools (reduction in their size), risk benefit ratio for off target effects.

Response: Thanks for your comment. The promising approaches to bypass these limitations have been mentioned in heading 6 accordingly. These modifications will improve the quality of the paper.

Comment 3: Major grammar and spell check is still required especially for gene and disease names (for example Line 127 ABCA4 not ABCD4)

Response: Thanks for highlighting it. The mentioned correction has been done in line 125 in the revised manuscript.

Comment 4: I think flow from one section from the other is completely missing, for example section 3 describes classical therapeutic approaches to restore vision while entering section 4 which describes use of CRISPR, section 3 should describe the limitations of classical therapeutic methods such as gene augmentation and why there is a need of alternative approaches.

Response: Thanks for your comment. I have edited section 3 (line number 213 to 216) accordingly.

Comment 5: what is the authors overall perspective of CRISPR genome editing in future ophthalmological clinical application. Where do author see CRISPR technology over the coming decade in terms of clinical translation rate when compared to other approaches. This should be described in the conclusion part. I think future perspective is missing in the last section.

Response: Thanks for your comments. The future perspective has been described accordingly from line number 760 to 768.

Round 3

Reviewer 1 Report

The Author has performed some revisions to the paper with the help of an ophthalmologist but some ophthalmological topics that are reported in the manuscript still need to be better defined, for example the fact that the production of Argus II has been suspended in favor of visual cortical prosthesis systems with the electrode neural interface moved from the retina to the visual cortex, or the fact that C2, CFB, and other gene variants were shown to be associated with polypoidal CNV.

In fact, the paragraph regarding genetic mutations associated with ocular diseases ought to be updated and improved.

Moreover, the text has still disjointed phrases and paragraphs that need further editing.

Author Response

Point-by-point response to the reviewer’s comments: 

Comment 1: The Author has performed some revisions to the paper with the help of an ophthalmologist but some ophthalmological topics that are reported in the manuscript still need to be better defined, for example the fact that the production of Argus II has been suspended in favor of visual corticalprosthesis systems with the electrode neural interface moved from the retina to the visual cortex, or the fact that C2, CFB, and other gene variants were shown to be associated with polypoidal CNV.

Response: Thanks for your comment. The manuscript has been reviewed by an ophthalmologist and acknowledged accordingly in the revised manuscript. The mentioned corrections regarding Argus II has been done (line 172-181) highlighted with Turquoise color in the revised manuscript. Moreover, the correction about gene variants C2 and CFB that have been revealed to be associated with polypoidal CNV has been done (line number 133-135) and its reference number 37 is also included in the revised manuscript.

Comment 2: In fact, the paragraph regarding genetic mutations associated with ocular diseases ought to be updated and improved.

Response: I am very thankful for the nice suggestion. In the first draft of the manuscript, reviewer 2 suggested to remove the complete section 2 “Genetic Causes of Retinal Degeneration and Blindness”. According to his comment “I would skip/ minimize all the info related to ocular genetic diseases and other therapeutic options as that is not the subject of the review”. Considering the mentioned comment and scope of the special issue of IJMS “Advances, Pitfalls and Future Perspectives for CRISPR/Cas9 Mediated Genome Editing, the author has tried to minimize this paragraph and has focused more on the application of CRISPR/Cas9 as a promising therapeutic tool to cure the blindness.

Comment 3: Moreover, the text has still disjointed phrases and paragraphs that need further editing.

Response: Thanks for your comment. The manuscript has been thoroughly checked for the grammatical mistakes/typos and edited it accordingly (line number 147-149). The new corrections are highlighted with Turquoise color in the revised manuscript.